# CENSORING REPRESENTATIONS WITH MULTIPLE-ADVERSARIES OVER RANDOM SUBSPACES

## ABSTRACT

Adversarial feature learning (AFL) is one of the promising ways for explicitly constrains neural networks to learn desired representations; for example, AFL could help to learn anonymized representations so as to avoid privacy issues. AFL learn such a representations by training the networks to deceive the adversary that predict the sensitive information from the network, and therefore, the success of the AFL heavily relies on the choice of the adversary. This paper proposes a novel design of the adversary, *multiple adversaries over random subspaces* (MARS) that instantiate the concept of the *volunerableness*. The proposed method is motivated by an assumption that deceiving an adversary could fail to give meaningful information if the adversary is easily fooled, and adversary rely on single classifier suffer from this issues. In contrast, the proposed method is designed to be less vulnerable, by utilizing the ensemble of independent classifiers where each classifier tries to predict sensitive variables from a different *subset* of the representations. The empirical validations on three user-anonymization tasks show that our proposed method achieves state-of-the-art performances in all three datasets without significantly harming the utility of data. This is significant because it gives new implications about designing the adversary, which is important to improve the performance of AFL.

## 1 INTRODUCTION

Since its invention over ten years ago (Hinton et al., 2006), deep neural networks (DNN) have shown significant performance improvements in various fields. When we apply DNN or more general machine learning techniques to real-world data, one of the key challenges is how to systematically incorporate the desired constraints into the learned representations in a controllable manner. For example, when practitioners apply these techniques to the data that contain a lot of user information (such as images with username (Edwards & Storkey, 2016) or data of wearables (Iwasawa et al., 2017)), the desired representations should not contain user-information that may result in privacy issues. Moreover, for legal and ethical reasons, machine learning algorithms have to make fair decisions, which do not rely on sensitive variables such as gender, age, or race (Louizos et al., 2016; Edwards & Storkey, 2016). Such a background requires removal of information related to specific factors (such as user ID, race, etc.) from the representation; this is called *censoring representations* in this paper.

One of the recently proposed approaches for censoring representation is *adversarial feature learning* (AFL) (Edwards & Storkey, 2016; Iwasawa et al., 2017; Xie et al., 2017), which employs the adversarial training framework to constrain the representations (Figure 1-a). Specifically, AFL considers an adversarial classifier who attempts to predict sensitive variables from the representations of a DNN and simultaneously trains the DNN to deceive the classifier. By alternatively or jointly (using gradient reversal layer proposed by Ganin & Lempitsky (2015)) training the adversary and DNN in such a manner, AFL ensures that there is little or no information about the sensitive variables in the representations.

Although some previous studies report significant performance improvements of the AFL in the context of censoring representations, the success of the AFL depends on the choice of the adversarial classifier. For example, if we use a logistic regression as the adversarial classifier, AFL can only eliminate the information that is linearly separated in the representation spaces and cannot remove

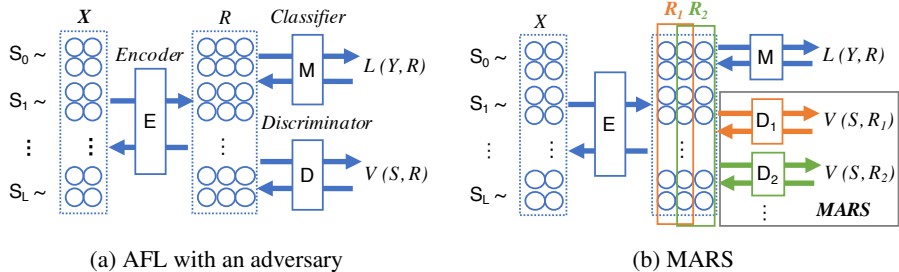

(a) AFL with an adversary                   (b) MARS

Figure 1: Multiple Adversaries over Random Subspaces (MARS) for censoring representations: (a)AFL with an adversary, (b) AFL with MARS. MARS is designed to have diverse-view and to be less vulnerable to meaningless update of the encoder $E$.

any non-linear dependency. It is also possible that deceiving some classifier might be too easy, resulting in poor performance improvements of AFL. As such, the design of adversary is crucial for the performance of AFL; however, existing studies fail to address how to design the adversary for improving the quality of AFL.

In this paper, we propose a novel design of adversary for improving the performance of AFL, *multiple-adversaries over random subspace* (MARS), which consider the *vulnerableness* of the adversary. The proposed design is motivated by the recent report (Iwasawa et al., 2017) that is just increasing the capacity of adversary did not successfully improves the performance of AFL (Iwasawa et al., 2017), and assumptions that deceiving an adversary fail to give meaningful information if the adversary is easily fooled, and adversary relies on single classifier suffer from this issues. The proposed method incorporates multiple adversaries where each adversary tries to predict sensitive variables from a different *subset* of the representations. This design makes adversary less vulnerable to the update of the encoder since the encoder needs to in a set of diverse adversaries. In this paper, we validate the effectiveness of the proposed design by empirically showing that (1) MARS archives better performance compared to baselines (that uses a single adversary and multiple adversaries over the entire representation spaces), and (2) MARS is less vulnerable compared to the baselines.

The primary contributions of this paper are as follows:

- This is the first study verifying the importance of the design of adversary in AFL and proposes the novel design for improving AFL. This is significant because the results suggest that design of adversary is vital for the performance of adversary, and gives new implications about designing the adversary in AFL, which is important to improve the performance of AFL. It is worth mentioning that, except our paper, all existing studies focus only on the *accuracy/capacity* for designing adversaries, which is not enough for improving the performance of AFL as shown in this paper.

- The proposed method achieved state-of-the-art performance in the task of censoring representations, which is essential to extend the applicability of DNN to many real-world applications. The empirical validation using three user-anonymization tasks shows that the proposed method allows the learning of significantly more anonymized representations with negligible performance degradation. Specifically, the probability of correctly predicting the user ID from learned representations is more than $0.07$ points better on average than that of a single adversary and multiple adversaries over entire representation spaces.

## 2    PROBLEM DEFINITION AND RELATED WORKS

### 2.1    PROBLEM DEFINITION: CENSORING REPRESENTATIONS

Censoring representation is a task to obtaining unbiased features. Here, *unbiased features* are features that are less affected by $S$, where $S$ is a random variable that we want to remove from the data for some reason. One typical reason is related to fairness or privacy, which requires the output of neural networks not to be affected by unfair information or not contain user information.

It should be noted that poor design of the censoring procedure significantly reduces the utility of data. For example, the output of random mapping $f_{rand}$ apparently has no information about $S$,

but it also gives no information about target $Y$. Alternatively, as a more realistic example, a neural network with limited capacity possibly acquires less information about $S$, but it may also result in poorer performance. Therefore, the primary goal of censoring representation is to obtain an encoder $E$ that reduces information about $S$, while maintaining information about $Y$. Formally, the task can be written as a joint optimization problem of the loss:

$$J = L(E(X), Y) - \lambda V(E(X), S),$$

where $X$ indicates the input random variable, $E$ is an encoder that transforms $X$ to representation $R$, $\lambda$ is the weighting parameter, and $V$ and $L$ are loss functions that represent how much information about $S$ and $Y$ is present, respectively. Note that $S$ can be any form of variables such as binary variable, categorical variable, or continuous variable. In this paper, we primarily consider a particular variant of censoring representation tasks, where we learn $E$ with deep neural networks and $S$ is the user ID (anonymization tasks).

## 2.2 RELATED WORKS: ADVERSARIAL FEATURE LEARNING

A recently proposed approach for the censoring representation task is *adversarial feature learning* (AFL). In AFL, $V(E(X), S)$ is measured by an external neural network $D$. In other words, if the external networks can accurately predict $S$ from $R$, AFL regards that $R$ has too much information about $S$ and if it is difficult to predict $S$ accurately, AFL regards that $R$ has little or no information about $S$. The external network $D$ is called discriminator or adversary in this context. The information is used to update the weights of the encoder $E$ so that the updated representations have less information about $S$. Formally, AFL solves the joint optimization problems:

$$\min_{E,M} \max_{D} \mathbb{E}[-\log q_M(y|h = E(x)) + \lambda \log q_D(s|h = E(x))],$$

where $M : R \to Y$ is the label classifier, $D : R \to S$ is the adversarial classifier (see Figure 1), and $q_M$ and $q_D$ are conditional distributions parameterized by $M$ and $D$, respectively. The optimization procedure is called adversarial training because $E$ and $D$ are in an adversarial relationship ($E$ tries to minimize of log-likelihood $q_D$ and $D$ tries to maximize it).

To the best of the authors' knowledge, the adversarial training framework was first introduced by (Schmidhuber, 1992), and later re-invented by (Goodfellow et al., 2014) in the context of an image-generation task. In the context of censoring representations, Edwards & Storkey (2016) first proposed the use of adversarial training to remove sensitive information from the representations. They show its efficacy for fair classification tasks ($S$ is binary) compared to an LFR (Learned Fair Representation) proposed by Zemel et al. (2013) that regularizes the $l_1$ distance between the distributions for data with different $S$. Iwasawa et al. (2017) first applied the adversarial training to learn anonymized representations on the data of wearables, where $S$ is categorical. More recently, Xie et al. (2017) first introduced the notion of *adversarial feature learning* and showed superior performances compared to variational fair auto-encoder (Louizos et al., 2016).

Although AFL already shows state-of-the-art performance to learn unbiased representations, how to improve its performance is still a challenge, which is tackled in this paper. This work is motivated by previous studies conducted by Iwasawa et al. (2017). They reported that simply using high-capacity networks as an adversary (such as deep neural networks) sometimes *fails* to improve the performance; on the other hand, the learned representations are still highly biased if the adversary does not have enough capacity. Our work can be regarded as an introduction of the new design consideration (i.e., vulnerableness) of adversary for improving the quality of AFL and validation of it by proposing a method that instantiates the concept, which is not done in the previous works.

It is worth mentioning that the concept of multiple adversaries itself is already proposed and verified in the context of the image generation with adversarial training (Durugkar et al., 2017), and this paper does not insist that using multiple adversaries is the primal novelty. From the methodological perspective, the proposed method (MARS) can be seen as an extension of the concept of multiple adversaries by introducing the concept of diversity, and we verify that this extension is essential in the context of censoring representations. Though the primary focus of this paper is limited to the context of adversarial feature learning, we further discuss the applicability of the proposed method on the other application of adversarial training, such as image/text generation (Goodfellow et al., 2014; Zhang et al., 2017) or domain adaptation(Ganin & Lempitsky, 2015); this is done at the end of this paper.

## 3 MULTIPLE-ADVERSARIES OVER RANDOM SUBSPACES

### 3.1 ENSEMBLE OF THE DIVERSE CLASSIFIERS AS ADVERSARY

The proposed method, *multiple-adversaries over random subspaces (MARS)*, considers multiple adversaries where each adversary is in charge of different subsets of features. The development of MARS is motivated by the assumption that the adversary should not be vulnerable and an ensemble of diverse classifiers make the adversary less vulnerable, resulting in the improved performance of AFL. The experiment parts of this paper empirically validate the importance of the design by showing that (1) MARS gives superior performances compared to the method with single adversary or that of simple ensemble of adversaries without enforcing adversaries to have diverse view, and (2) using ensemble of independent classifiers as adversary actually makes the adversary robust.

It is worth mentioning that the proposed method could be benefited from the variance reduction because the ensemble of diverse classifiers typically reduces the variance. More generally, this technique is so-called random subspace method in the context of ensemble researches and widely used in a various methods, such as random forest (Ho, 1998). To make the what parts of the proposed method contribute to the performance gain clearer, we also verify the effect of variance reductions later in the experimental parts.

### 3.2 ALTHORIGHM

Figure 1 shows the comparison of overall architectures of between AFL with an adversary, and with multiple adversaries over random subspaces. Since the primary difference between naive AFL and MARS is how to build the adversary (training adversaries) and how to use the information of adversary (training encoders), we focus on the explanation about the part.

There are many possible options to select a subspace; however, this paper considers the case where each subspace is randomly selected. Suppose that $n$ is the number of dimensions of representation $R \in \mathbb{R}^n$, and $K$ is the number of adversaries. In MARS, each adversary $D_k$ is trained to predict $S$ over a randomly selected subset of features $R^k$, whose dimension is $m_k$ and $m_k < n$. Each adversary is trained to maximize the expected log-likelihood. Formally, the optimization problem for $D_k$ is as follows

$$\max_{\theta_{D_k}} \mathbb{E}[\log q_{D_k}(s|h_k = Sub_k(E(x)))], \tag{1}$$

where $Sub_k$ is a function that return the subset of $R$, which is fixed before the training. Precisely, each $Sub_k$ determine whether to remove the $i$-th dimension of $R$ with the probability of $\alpha$. The $\theta_{D_k}$ is the parameter of $D_k$. The optimization is done by stochastic gradient descent as usual.

The adversaries are then integrated, and a single prediction is made to train the encoder. Formally, the probability distribution of $q_D(s = s_i|h = E(x))$ is parameterized by an ensembled adversary $D$:

$$q_D(s = s_i|h = E(x)) = \frac{1}{K} \sum_{k}^{K} q_{D_k}(s|h_k), \tag{2}$$

which means, the predictions of adversaries $D_k$ are integrated by averaging the predictions. Note that integration operator $F$ is not essential to be the averaging, it may be the max operator or the soften version of the max operator proposed in (Durugkar et al., 2017).

The encoder is trained for minimizing the log-likelihood of Eq.2 and negative log-likelihood $q_M(y|h)$. Formally,

$$\min_{\theta_E, \theta_M} \mathbb{E}[-\log q_M(y|h) + \lambda \log \frac{1}{K} \sum_{k}^{K} q_{D_k}(s|h_k)], \tag{3}$$

where $\theta_E$ and $\theta_M$ are the parameter of $E$ and $M$ respectively. Algorithm 1 show overall algorithm.

## 4 EXPERIMENTS

### 4.1 DATASETS

We used two datasets to demonstrate the efficacy of MARS. Both datasets are related to human activity recognition using the data of wearables, where privacy issues have recently been pointed

---

**Algorithm 1** Optimization of the proposed model

---

**Require:** parameter $\lambda$, $\alpha$
   Initialize $Sub_k$ for all $k \in 1 \cdots K$ based on $\alpha$
   Initialize neural networks $\{E, M, D_1, \cdots D_K\}$
   **while** training() **do**
      Update weights of $D_k$ for all $k \in 1 \cdots K$ (eq. 1)
      Update weights of $E$ and $M$ (eq. 3)
   **end while**

---

out by (Iwasawa et al., 2017). In both datasets, the task is to learn anonymized representations ($R$ that does not contain user-identifiable information), while maintaining classification performance.

The opportunity recognition dataset (Sagha et al., 2011) consists of sensory data regarding human activity in a breakfast scenario, and the task is to predict the activity performed by a user. A wide variety of body-worn, object-based, and ambient sensors are used (see Figure 1 in Sagha et al. (2011) for more details). Each record consists of 113 real-value sensory readings, excluding time information. We considered two recognition tasks: gesture recognition (Opp-G) and locomotion recognition (Opp-L). Opp-G requires the recognition of 18 class activities [1], while Opp-L requires the recognition of 4 class locomotion-activities, i.e., stand, walk, sit, and lie. Given a sampling frequency of 30 Hz, the sliding window procedure with 30 frames and a 50% overlap produced 57,790 samples. We used data from subjects 2–4 as training/validation sets (90% for training and 10% for validation) and that of subject 1 as test sets.

The USC-HAD dataset (Zhang & Sawchuk, 2012) is a relatively new benchmark dataset, which contains a relatively large number of subjects (14 subjects: 7 males and 7 females). The data include 12 activity classes that are corresponding to the most basic and everyday activities of people's daily lives [2]. MotionNode, which is a 6-DOF inertial measurement unit specially designed for human motion sensing applications, is used to record the outputs from accelerometers that record 6 real sensory values. The sliding window procedure, using 30 frames and a 50% overlap, produced 172,169 samples. We used data from subjects 1–10 as training/validation sets (90% for training and 10% for validation) and that of subjects 11–14 as test sets.

## 4.2 Experimental Setting

In all experiments, we parameterized the encoder $E$ by convolutional neural networks (CNN) with three convolution-ReLU-pooling repeats followed by one fully connected layer and $M$ by logistic regression, following a previous study Iwasawa et al. (2017). Every component of the network was trained with the Adam algorithm (learning rate is set to 0.0001) (Kingma & Ba, 2015) (150 epochs). In the optimization, we used the annealing heuristics for weighting parameter $\lambda$ following (Iwasawa et al., 2017); we linearly increased $\lambda$ during 15 epochs to 135 epochs of training.

Following previous works, we evaluated the level of anonymization by training a classifier $f_{eva}$ that tries to predict $S$ over learned representations. This means that we regarded that a learned representation is anonymized well if we could not build an accurate classifier. To be more specific, we trained the evaluator with the data that is used for training encoder $E$ and a part of the data whose data did not use for training the encoder $E$. The rest of the data is used for the evaluations.

To demonstrate the efficacy of the proposed method, we compare the following four methods and its variants: (1) None: w/o adversary (correspond to standard CNN), (2) Adv: w/ a single adversary, (3) MA: w/ multiple adversaries where each adversary tries to predict from the entire space, and (4) MARS: w/ multiple adversaries where each adversary tries to predict from a different subspace of the entire space. Each adversary is parametrized by multi-layer perceptron (MLP) with 800 hidden units. If we need to express the type of adversary, we denote it by a suffix. For example, Adv-LR means logistic regression parametrizes an adversary. Without mentioning otherwise, we set the num-

---

[1] open door 1, open door 2, close door 1, close door 2, open fridge, close fridge, open dishwasher, close dishwasher, open drawer 1, close drawer 1, open drawer 2, close drawer 2, open drawer 3, close drawer 3, clean table, drink from cup, toggle switch, and Null

[2] walking forward, walking left, walking right, walking upstairs, walking downstairs, running forward, jumping, sitting, standing, sleeping, elevator up, elevator down

Table 1: Performance comparison against various $f_{eva}$: (a, b, c) Accuracy on predicting $s$ in Opp-G, Opp-L, and USC-HAD datasets respectively, and (d)Accuracy on predicting $y$ in each dataset.

(a) Opp-G

|  | LR | MLP$_1$ | MLP$_2$ | DNN |
|---:|---|---|---|---|
| None | 0.806 | 0.936 | 0.971 | 0.968 |
| Adv-LR | 0.513 | 0.934 | 0.970 | 0.965 |
| Adv | **0.479** | 0.676 | 0.865 | 0.960 |
| Adv$_2$ | **0.472** | 0.672 | 0.853 | 0.955 |
| Adv$_5$ | 0.602 | 0.842 | 0.939 | 0.933 |
| Adv-DNN | 0.495 | 0.689 | 0.900 | 0.903 |
| MA | 0.571 | 0.782 | 0.922 | 0.942 |
| MA-DNN | 0.526 | 0.755 | 0.924 | 0.962 |
| MARS | 0.494 | **0.589** | **0.777** | **0.817** |
| MARS-DNN | 0.546 | **0.636** | **0.835** | **0.820** |

(b) Opp-L

|  | LR | MLP$_1$ | MLP$_2$ | DNN |
|---:|---|---|---|---|
| None | 0.839 | 0.937 | 0.970 | 0.966 |
| Adv-LR | 0.583 | 0.925 | 0.971 | 0.964 |
| Adv | **0.481** | **0.615** | 0.805 | 0.932 |
| Adv$_2$ | 0.593 | 0.811 | 0.934 | 0.928 |
| Adv$_5$ | 0.610 | 0.775 | 0.910 | 0.910 |
| Adv-DNN | **0.484** | 0.712 | 0.879 | 0.945 |
| MA | 0.527 | 0.744 | 0.905 | 0.949 |
| MA-DNN | 0.541 | 0.726 | 0.873 | 0.904 |
| MARS | 0.515 | 0.638 | **0.791** | **0.850** |
| MARS-DNN | 0.500 | **0.615** | **0.781** | **0.866** |

(c) USC-HAD

|  | LR | MLP$_1$ | MLP$_2$ | DNN |
|---:|---|---|---|---|
| None | 0.655 | 0.741 | 0.809 | 0.808 |
| Adv-LR | **0.275** | 0.511 | 0.721 | 0.774 |
| Adv | 0.325 | 0.513 | 0.703 | 0.762 |
| Adv$_2$ | 0.538 | 0.662 | 0.779 | 0.801 |
| Adv$_5$ | 0.377 | 0.647 | 0.771 | 0.792 |
| Adv-DNN | **0.295** | 0.499 | **0.668** | 0.742 |
| MA | 0.326 | 0.603 | 0.743 | 0.783 |
| MA-DNN | 0.362 | **0.497** | 0.672 | **0.740** |
| MARS | 0.354 | **0.482** | 0.692 | 0.751 |
| MARS-DNN | 0.364 | 0.498 | **0.655** | **0.680** |

(d) Accuracy on predicting $y$

|  | Opp-G | Opp-L | USC | Ave |
|---:|---|---|---|---|
| None | 0.796 | 0.828 | 0.537 | 0.720 |
| Adv-LR | 0.805 | 0.820 | 0.535 | 0.720 |
| Adv | 0.805 | 0.808 | 0.540 | 0.718 |
| Adv$_2$ | 0.803 | 0.823 | 0.521 | 0.716 |
| Adv$_5$ | 0.805 | 0.827 | 0.529 | 0.721 |
| Adv-DNN | 0.804 | 0.797 | 0.532 | 0.711 |
| MA | 0.802 | 0.792 | 0.528 | 0.708 |
| MA-DNN | 0.776 | 0.791 | 0.513 | 0.694 |
| MARS | 0.802 | 0.824 | 0.528 | 0.718 |
| MARS-DNN | 0.804 | 0.815 | 0.511 | 0.710 |

ber of adversaries $K$ as 10 for MA and MARS, and $\alpha$ for determine the size of the subset as 0.8. As expected from equation 3, the selection of weighting hyper-parameter $\lambda$ greatly affects the performances. Without mentioning otherwise, we optimized the $\lambda$ from $\{0.01, 0.02, 0.05, 0.10, 0.20, 1.0\}$ for each baseline for the fair comparison. Specifically, we select the $\lambda$ that did *not* significantly harm $Y$-accuracy (specifically, 5% or less degradation against None) and maximize the level of anonymization. The selected $\lambda$ for each baseline is listed in appendix A.

## 4.3 EXPERIMENTAL RESULTS

Table 1-a,b, and c list the user classification accuracy for the three tasks with different adversarial classifiers $D$ and evaluator $f_{eva}$. Table 1-d shows the label classification accuracy for each datasets. In addition to None, Adv, MA, MARS, we compared a logistic repression (Adv-LR), DNNs with 400-200 hidden units, the ensemble of DNNs (MA-DNN), and the ensemble of DNNs over random subspaces (MARS-DNN) for $E$. Adv$_2$ and Adv$_5$ correspond to the case where we train an adversary with MLP 2 iterations or 5 iterations against one iteration for training encoder $E$. For evaluator $f_{eva}$, we tested LR, multi-layer perceptron with 50 hidden units (MLP$_1$), multi-layer perceptron with 800 hidden units (MLP$_2$), and deep neural networks with 400-200 hidden units. The best performance in each combination of datasets and $f_{eva}$ is highlighted in underline and bold. The second best performance is highlighted in bold.

We can make the following observations. (1) For small-capacity evaluator (LR), a single adversary (Adv and its variants) marks superior performances to MA and MARS. However, the method with single adversary tends to give poor performance for the stronger adversary (MLP$_2$ or DNN). (2) For high-capacity evaluator (MLP$_2$ or DNN), the proposed method outperforms baseline methods. Specifically, in the case when the evaluator is DNN, MARS and MARS-DNN marks 0.806 and 0.789 on average of user accuracy, which is significantly better than the best baseline Adv-DNN (0.863 on average of user accuracy). (3) Increasing the iterations of training adversary does not pay off, as shown in the weak performance improvements of Adv$_2$ or Adv$_5$ against Adv.

In addition to the quantitative evaluation, we also give qualitative evaluation by visualizing the learned representations using t-SNE (Maaten & Hinton, 2008) (Figure 2). Precisely, we randomly sample 1,000 examples from the validation sets and reduce the dimension of data by using principal component analysis over the output of the $E$ to speed up the computations and suppress some noise. The number of components is selected such that the amount of variance that needs to be explained

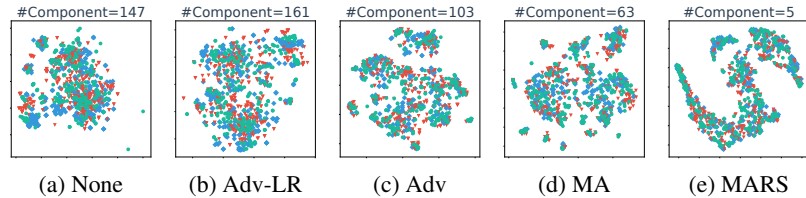

Figure 2: t-SNE visualizations of samples in the Opp-G (the original dimension is 400).

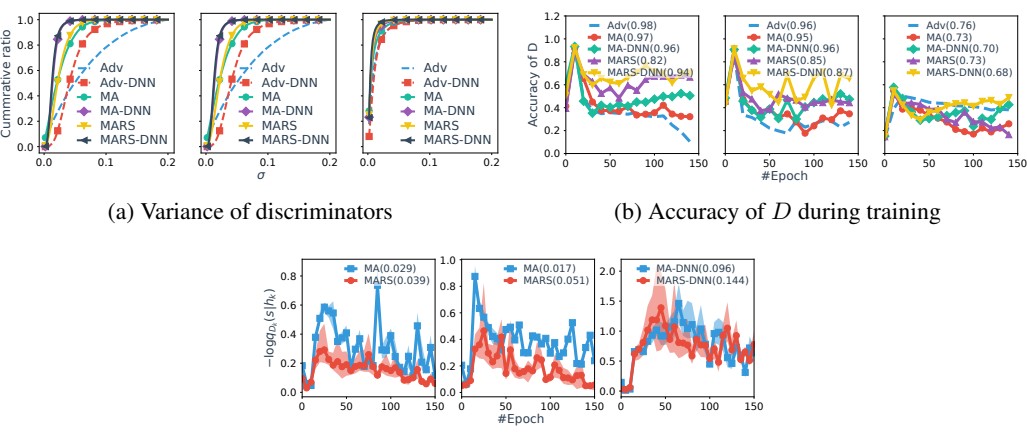

is higher than $95\%$. The selected number of components is mentioned on top of each figure. Each color (and marker) denotes different users ($|S|$=3). The results show that the representations learned with MARS provide qualitatively more plausive results; we can see that the data for different users lie in almost similar places in the representations of MARS; however, the representations of None, Adv-LR, and Adv, MA are partially clustered based on the users. Moreover, the representation of MARS a significantly small number of components to represent the leaned information, which possibly suggests that MARS eliminates unimportant information to a greater extent.

Figure 3-a compares the variance of baselines and the proposed methods. To be more specific, we calculated the variance of each adversary by training the classifier five times, and compute the expected variance of the activations: $\mathbb{E}_{h=E(x)\in X}[\sigma(D'(h))]$, where $D'$ denote the output of the adversary before Softmax non-lineality. The results show that MARS indeed have less variance compared to Adv, however, MARS and MA the almost same variance although MARS gives better performance compared to MA as shown in Table 1, suggesting that the effect of variance reduction is not main advantage of the proposed method.

Figure 3-b compares the accuracy of $D$ during training. The number listed next to each method denotes the final user accuracy measured by DNN. The interesting thing here is, one method fails to achieve good final performance and the other achieve it even if both methods give the similar accuracy of $D$ during training. For example, in the Opp-L dataset (center of the Figure), MA-DNN and MARS show similar performance curve during training, but MARS gives more than 0.10 better score at final (0.85 to 0.96). This result implies that accuracy of the discriminator is not an only definitive factor that determines the success of AFL.

Figure 3-c shows how the update of $E$ affect the $-\log q_{D_k}$. Specifically, each line represents the average effect of the loss by the update $E$, and the shaded area represents the minimum and maximum effect of the loss, which means that larger shaded area indicates the higher variety of the $D_k$. The number listed next to each method represents the standard deviation of the classifiers. The result demonstrates that the influences of the update to each adversary are more varied for MARS compared to MA, which implies the adversaries of higher $\alpha$ have more diversity, and it makes the $D$ less vulnerable against the update.

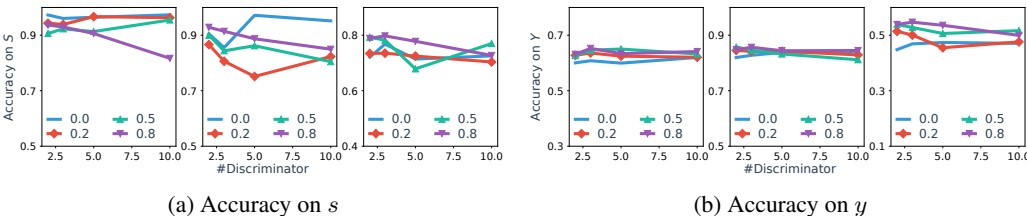

(a) Accuracy on $s$             (b) Accuracy on $y$

Figure 4: Predicting accuracy on (a) $s$ and (b) $y$ with different $K$ and $\alpha$ in each dataset: (Left) Opp-G, (Center) Opp-L, (Right) USC.

Figures 4-a and b compares the performance between different $K$ and $\alpha$. Note that $\alpha = 0.0$ means the method using all feature spaces, which is MA. The results shows that (1) although there is no clear winner among the $\alpha$, using subspaces gives lower accuracy on predicting $S$, (2) the number of discriminator $K$ effect accuracy on $S$ while the accuracy on $Y$ is robust to the $K$ especially if $\alpha \neq 0$, indicating the possibility that the performance of AFL could be improved by incorporating multiple adversaries.

## 5    DISCUSSION AND CONCLUSION

This study proposed MARS, which incorporates multiple adversaries where each adversary has a different role and conducted empirical validations on the efficacy of the proposed method for censoring representations, specifically user-anonymization for the data of wearables. Table 1 compares the proposed method and several baselines and shows the efficacy of the proposed method against various evaluators. Figure 2 qualitatively shows that the proposed method provides well-anonymized representations. Figure 3-c shows that each adversary in MARS has the diverse role, resulting MARS more robust to the update of $E$ as a whole. All these results support that the proposed method is more effective in removing the influence of a specific factor (user in experiments) compared to the previous methods.

One of the reasons why MARS works well is that the adversary is designed to have diverse-views by incorporating random subspace methods, resulting the encoder need to be stronger to deceive the adversary. It is worth mentioning that the capacity or accuracy of the adversary is not the only a definitive factor that determines the success of the adversarial feature learning, as shown by the superior performance of MARS over MA that has $\frac{1}{1-\alpha}$ times the larger capacity of MARS. Moreover, the final performance of AFL is significantly different even if the accuracy of $D$ is reasonably similar during training, as shown in Figure 3-b. As mentioned in the related work section, such knowledge is essential to design the adversary in practice, and prior studies of adversarial feature learning did not address this issues.

Although this paper focused on the case where the subsets are randomly selected and fixed, this might not be necessary. One of the possible extensions is to determine subsets with more sophisticated ways (e.g., by performing clustering or soft-clustering on the representation spaces after few training iterations), or to learn how to select the subset itself by adding the criterion regarding the diversity of adversaries. Also, it might be possible to realize the diversity of adversaries by methods other than subspace selection. One possible way is to constrain weights of two adversaries so that they are an orthogonal view, which is used in semi-supervised learning using co-training (Saito et al., 2017), or it might be worth a try to add different noises for each adversary.

It might be worth mentioning about applicability and implications of MARS for other applications of adversarial training, such as image generation. From the perspective of the applicability, the MARS itself does not rely on any domain-specific settings and is therefore general enough for many applications based on adversarial training. For example, we can build multiple-adversaries upon the subset of feature spaces (maybe not on the image spaces). This makes discriminator have diverse-view, so it might be useful for preventing *mode collapse* that is one of the well-known problems in image-generation with adversarial training. In the context of image-generation, Generative Multi Adversarial Networks proposed by Durugkar et al. (2017), which also use multiple adversaries, shows that multiple adversaries are useful for generating better images, and for avoiding mode collapse. It might be interesting to see if enhancing the diversity of discriminators by preparing asymmetric adversaries as with this paper helps to generate a better image or to avoid mode collapse better.

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

## A    SELECTED HYPER PARAMETER

Table 2 shows the selected $\lambda$ for each combination of datasets and baselines. Although the best hyper-parameter might be determined by the balance between $\log q_M$ and $\log q_D$, here we cannot see the obvious relationships between the best $\lambda$ and the easiness of tasks.

Table 2: $\lambda$ used in Table 1 and Figure 2.

|          | OppG  | OppL  | USC   |
|---------:|-------|-------|-------|
| Adv-LR   | 0.100 | 0.200 | 1.000 |
| Adv      | 0.200 | 0.050 | 0.100 |
| $Adv_2$  | 0.100 | 0.010 | 0.010 |
| $Adv_5$  | 0.010 | 0.010 | 0.020 |
| Adv-DNN  | 0.100 | 0.050 | 0.020 |
| MA       | 0.200 | 0.200 | 0.020 |
| MA-DNN   | 1.000 | 0.200 | 0.050 |
| MARS     | 1.000 | 1.000 | 0.050 |
| MARS-DNN | 0.200 | 1.000 | 0.100 |

