# OpenReview forum: "Censoring Representations with Multiple-Adversaries over Random Subspaces"
_ICLR.cc/2018/Conference — Reject_

### Official Review · AnonReviewer1 · 2017-11-27
**Improved performance in obtaining censoring representations, but contribution not quite signficant enough**

**Rating:** 5
**Confidence:** 4

**Review:**

- The authors propose the use of multiple adversaries over random subspaces of features in adversarial feature learning to produce censoring representations. They show that their idea is effective in reducing private information leakage, but this idea alone might not be signifcant enough as a contribution.

- The idea of training multiple adversaries over random subspaces is very similar to the idea of random forests which help with variance reduction. Indeed judging from the large variance in the accuracy of predicting S in Table 1a-c for single adversaries, I suspect one of the main advantage of the current MARS method comes from variance reduction. The author also mentioned using high capacity networks as adversaries does not work well in practice in the introduction, and this could also be due to the high model variance of such high capacity networks.

- The definition of S, the private information set, is not clear. There is no statement about it in the experiments section, and I assume S is the subject identity. But this makes the train-test split described in 4.1 rather odd, since there is no overlap of subjects in the train-test split. We need clarifications on these experimental details.

- Judging from Figure 2 and Table 1, all the methods tested are not effective in hiding the private information S in the learned representation. Even though the proposed method works better, the prediction accuracies of S are still high.

---

> ### Author Response · Authors · 2017-12-04
> **Reply for reviewer1**
>
> Thank you for reading and commenting our paper.
>
> > The idea of training multiple adversaries over random subspaces is very similar to the idea of random forests which help with variance reduction.
>
> Thank you for the valuable comment. As the reviewer's comment, MARS is similar to random forest in surface (actually both methods could be regarded as the instantiation of more general method "random subspace methods"), and it is possible to some parts of the success come from variance reduction property, as with random forests.
>
> However, we are not fully agree that the main advantage of the MARS comes from variance reduction, based on the observations: (1) In the first place, the training of the discriminator(s) are {\em not} suffered from large variance. Specifically, in the task of predicting $s$, almost no performance differences is observed between two subsets of datasets (training and validation) throughout the training. We will add some experimental results in the Appendix parts (2) As shown in figure 4-c, the classification performance of different hyper-parameter $\alpha$ are almost equally raise at the beginning of training. Note that, $\alpha=0.0$ corresponds to the MA, and $\alpha=0.8$ corresponds to the MARS in the Table 1. Since the large variance make the training slower typically, it imply that the superior of the MARS to MA seems not come from only the variance reduction parts.
>
> Rather, our results suggests that the usage of random subspaces make it hard to deceive discriminators (indicated by the high variance in figure 4-d). This is intuitive results as the encoder need to deceive discriminators that have various views, rather than the single view point. This makes the encoder need to be more stronger to beat the discriminators, and make the representations more invariant to the sensitive variable $S$ (as shown in Table1).  This is what we ephasized as “vulnerableness” in the current manuscript.
>
> Any way, we will definitely add some explanations about this topic (may be with making new subsection at the end of section 3). Again, thanks for the valuable reviews.
>
>
> > The definition of S, the private information set, is not clear. There is no statement about it in the experiments section, and I assume S is the subject identity. But this makes the train-test split described in 4.1 rather odd, since there is no overlap of subjects in the train-test split. We need clarifications on these experimental details.
>
> As you assumption, the definition of $S$ is user identity. I will make it clearer this in the final manuscript.
>
> I’m sorry for confusing about the experimental settings. Actually, the performance about predicting $S$ is measured by different train-test split. Specifically, all training datasets and a part of test datasets (including new users) are used for train the evaluator $f_{eva}$, and the rest of the dataset is used for the evaluations.
>
>
> > Judging from Figure 2 and Table 1, all the methods tested are not effective in hiding the private information S in the learned representation. Even though the proposed method works better, the prediction accuracies of S are still high.
>
> This is true that even the proposed method could not effective enough to practical usages especially if considering strong adversaries; however, always do so with science. Moreover, evaluation itself is too conservative to discuss about the practical usages because evaluator have too much labeled datasets. Considering the importance of the task itself, we believe that proposing the new method that achieves state-of-the-art performance on this task is enough to the contributions. We’d like to emphasise this point again at the final manuscript.

---

### Official Review · AnonReviewer3 · 2017-11-28
**Multiple adversaries for increased privacy performs well**

**Rating:** 6
**Confidence:** 4

**Review:**

MARS is suggested to combine multiple adversaries with different roles.
Experiments show that it is suited to create censoring representations for increased anonymisation of data in the context of wearables.

Experiments a are satisfying and show good performance when compared to other methods.

It could be made clearer how significance is tested given the frequent usage of the term.

The idea is slightly novel, and the framework otherwise state-of-the-art.

The paper is well written, but can use some proof-reading.

Referencing is okay.

---

> ### Author Response · Authors · 2017-12-04
> **Reply for reviewer3**
>
> Quick Response for AnonnReview3
>
> Thank you for reading and commenting our paper.
>
> > It could be made clearer how significance is tested given the frequent usage of the term.
>
> I'm not sure I'm parsing your suggestion correctly. Are you suggesting to turn on/off the reguralization term for each epoch, or to compare the different number of adversaries? I think I could give more specific answer if you could give a bit detail about your suggestion and motivation behind it.
>
> > The paper is well written, but can use some proof-reading.
>
> We will definitely update our manuscript with proof-reading. Also, we could give some explanation if you could give us a specific parts you are hard to understand.
>
>
> We look forward to hearing from you regarding our submission. We would be glad to respond to any further questions and comments that you may have.

---

### Official Review · AnonReviewer2 · 2017-12-02
**Minor theoretical contribution; main focus on experiments**

**Rating:** 6
**Confidence:** 3

**Review:**

The below review addresses the first revision of the paper. The revised version does address my concerns. The fact that the paper does not come with substantial theoretical contributions/justification still stands out.

---

The authors present a variant of the adversarial feature learning (AFL) approach by Edwards & Storkey. AFL aims to find a data representation that allows to construct a predictive model for target variable Y, and at the same time prevents to build a predictor for sensitive variable S. The key idea is to solve a minimax problem where the log-likelihood of a model predicting Y is maximized, and the log-likelihood of an adversarial model predicting S is minimized. The authors suggest the use of multiple adversarial models, which can be interpreted as using an ensemble model instead of a single model.

The way the log-likelihoods of the multiple adversarial models are aggregated does not yield a probability distribution as stated in Eq. 2. While there is no requirement to have a distribution here - a simple loss term is sufficient - the scale of this term differs compared to calibrated log-likelihoods coming from a single adversary. Hence, lambda in Eq. 3 may need to be chosen differently depending on the adversarial model. Without tuning lambda for each method, the empirical experiments seem unfair. This may also explain why, for example, the baseline method with one adversary effectively fails for Opp-L. A better comparison would be to plot the performance of the predictor of S against the performance of Y for varying lambdas. The area under this curve allows much better to compare the various methods.

There are little theoretical contributions. Basically, instead of a single adversarial model - e.g., a single-layer NN or a multi-layer NN - the authors propose to train multiple adversarial models on different views of the data. An alternative interpretation is to use an ensemble learner where each learner is trained on a different (overlapping) feature set. Though, there is no theoretical justification why ensemble learning is expected to better trade-off model capacity and robustness against an adversary. Tuning the architecture of the single multi-layer NN adversary might be as good?

In short, in the current experiments, the trade-off of the predictive performance and the effectiveness of obtaining anonymized representations effectively differs between the compared methods. This renders the comparison unfair. Given that there is also no theoretical argument why an ensemble approach is expected to perform better, I recommend to reject the paper.

---

> ### Author Response · Authors · 2017-12-04
> **Reply for reviewer2**
>
> Thank you for reading and commenting our paper.
>
> > The way the log-likelihoods of the multiple adversarial models are aggregated does not yield a probability distribution as stated in Eq. 2. While there is no requirement to have a distribution here - a simple loss term is sufficient - the scale of this term differs compared to calibrated log-likelihoods coming from a single adversary. Hence, lambda in Eq. 3 may need to be chosen differently depending on the adversarial model. Without tuning lambda for each method, the empirical experiments seem unfair. This may also explain why, for example, the baseline method with one adversary effectively fails for Opp-L. A better comparison would be to plot the performance of the predictor of S against the performance of Y for varying lambdas. The area under this curve allows much better to compare the various methods.
>
> First of all, I’m sorry for confusing you about the experimental settings. Although we have mentioned that “ The hyper-parameter is set to 1.0, 1.0, 0.1 for Opp-G, Opp-L, and USC-HAD respectively”, these are {\em not fixed} thorough out the experiments. We tuned the $lambda$ of baselines ($Adv_{0.1}$ $MA_{0.1}$ in Table 2). It indeed increase the performances; however the proposed method still outperforms the baselines. Note that, we also tested the case where $\lambda = {0.01, 0.1, 0.2, 1.0}$, but the results are consistent,. though we have removed it due to the tight space limitation.  We’d like to clarify this points and add some experimental results with different $\lambda$ in the final manuscript.
>
> Secondly, I’m afraid there is some misunderstanding regarding the Eq (2). It just averaging the $q_{D_k}$.  As the each $q_{D_k}$ is  calibrated, the averaged model is also calibrated (the sum of $q_D$ is 1.0). Although it is true that model averaging possibly tend to give smooth values and it make log-likelihood little bit differ, I don’t think it needs special treatments except testing different hyper-parameters, as mentioned above. I’m appreciate if the reviewer could clarify the concern.
>
> > Though, there is no theoretical justification why ensemble learning is expected to better trade-off model capacity and robustness against an adversary. Tuning the architecture of the single multi-layer NN adversary might be as good?
>
> As the reviewer’s mentioned (and also described in the end of the introduction of our paper), the main contributions of this paper come from empirical validations of the superior performance of the proposed method compared to various baselines tested on various hyper parameters (e.g., Table 2). This is significant because it shade a light on new design consideration for improving the performance of the AFL framework, as mentioned in abstract and introduction of our paper. Considering the importance of the task, we believe introducing the new design consideration and the state-of-the-art method are enough contributions.
>
>
> Thanks again for reading of our paper.

---

> > ### Comment · AnonReviewer2 · 2017-12-04
> > **Please add your additional experiment results**
> >
> > In the paper, lambda is chosen differently for the different datasets, but not for the methods. However, lambda effects each method differently as it balances different terms for the various models (unless all models return calibrated probability distributions). If you have experiments for various lambdas, I would encourage you to add those to the paper. One way - without running into space limitations - would be to search for the maximal lambda (for each method separately) so that the method still meets a certain fixed performance on Y, and then report the performance for S. Similar, for example, as you fix recall and report precision. If your method still performs better, I'm more than welcome to change my vote.
> >
> > The averaged likelihoods (not avg. log-likelihoods) do not form a likelihood anymore as you cannot normalize it independently of the model. Again, this is not a problem; for example, the SVM's hinge loss doesn't yield a proper log-likelihood either. The problem is that because of this, lambda has a different impact on how you trade-off the performances on Y and S. And there are good chances that this difference is significant: S seems to be easy to predict from the full representation (close to 100% accuracy for Opp-G and Opp-L). The log-likelihood term (weighted by lambda) is apparently close to 0 for the single adversary (which is also a result of the choice of lambda). I would expect that the averaged likelihood is much higher, simply because you will likely have at least one feature subset, which does not reveal S and the loss will be not zero. Hence, for the same lambda, the averaged likelihood would be higher and the optimizer has a higher incentive to change the representation.

---

### Author Response · Authors · 2018-01-05
**We have updated our manuscript. Main revisions are (1) Hyper-parameter optimization for baselines, (2) Analysis on the effect of variance reduction, (3) Clarifying the significance/contributions.**

Following reviewer's comments and concerns, we have revised our paper.

# Major revisions (three points)
(1) Table 1 and Figure 2. We replace the results of each baseline by optimizing hyper-parameter $lambda$ for each baseline (following reviewer 2's concern). The results show that the proposed method still achieves performance improvements compared to baselines (w/ single adversary and w/ multiple adversaries over entire feature spaces). Please see the Table 1, Figure 2, and related paragraphs for more detail. The procedure for the hyperparameter selection is described at the end of section 4.1.
Note that, the original version of the manuscript also shows the performance with $\lambda$ varied for some baselines ($Adv_{0.1}, MA_{0.1}$), so this revision is not entirely new in the final manuscript.

(2) Figure 3. We added analysis of the effect of variance reduction (following reviewer 1's comment).
In response to the comment of review 1, we compared the effect of variance reduction among baselines and the proposed method. The results show that MARS tend to give superior performance to MA although both methods have almost similar variance, suggesting that the variance-reduction is not main advantage of the proposed method. Please see the Figure 3-a for more detail. We also add the proper explanation to the beginning of section 3.1.
We also add analysis on relationships between accuracy of discriminators and the final performance of AFL, to clarify key factor for the success of AFL. The results support that our underlying assumption, i.e., the capacity/accuracy of the adversary is not the dominant factor. Please refer the Figure 3-b for more detail.

(3) Abstract and paragraph 3--5 in Introduction.
As we have mentioned in the responses for reviewer's feedbacks, the primary contribution of our paper is (1) we proposed the novel design of adversary for AFL, and (2) it achieved state-of-the-art performance in several tasks related to the censoring representations. This is significant because the results shed light on the importance of the design of adversary, and gives new implications about the design. It is worth mentioning that, except our paper, all existing studies focus only on the {\em accuracy/capacity} for designing adversaries, which is not enough for improving the performance of AFL as shown in this paper. Moreover, the task itself is essential for using the power of Deep Neural Networks in many real-world applications, as mentioned in the introduction of the manuscript.
We have revised abstract and paragraph 3--5 of the Introduction based on the above discussion. We hope this revision makes the contributions clearer.

# Minor Revisions
- Delete the Figure 2 in the original manuscript (since it overlaps with Table 1 to some extent).
- Fix the grammatical and typographical mistakes
- Add some detail about the experimental setting, following the question of review 1.

---

### Decision · Program_Chairs · 2018-01-29
**ICLR 2018 Conference Acceptance Decision**

**Decision:**

Reject

**Comment:**

The reviewers tend to agree that the empirical results in this paper are good compared to the baselines. However, the paper in its current form is considered a bit too incremental. Some reviewers also suggested additional theory could help strengthen the paper.